# Simultaneous Inhibition of PD-1 and Stimulation of CD40 Signaling Pathways by Anti-PD-L1/CD40L Bispecific Fusion Protein Synergistically Activate Target and Effector Cells

**DOI:** 10.3390/ijms222111302

**Published:** 2021-10-21

**Authors:** Madhu S. Pandey, Chunlei Wang, Scott Umlauf, Shihua Lin

**Affiliations:** 1Bioassay Development, Analytical Sciences, Biopharmaceutical Development, R&D, AstraZeneca, Gaithersburg, MD 20878, USA; madhu.pandey@astrazeneca.com (M.S.P.); scott.umlauf@astrazeneca.com (S.U.); 2Physicochemical-Analytical Sciences, Biopharmaceutical Development, AstraZeneca, R&D, Gaithersburg, MD 20878, USA; chunlei.wang1@astrazeneca.com

**Keywords:** bioassay, bispecific antibody, potency, mechanism of action, qualification

## Abstract

Bispecific antibodies (BsAbs) or fusion proteins (BsAbFPs) present a promising strategy for cancer immunotherapy. Numerous BsAbs targeting coinhibitory and costimulatory pathways have been developed for retargeting T cells and antigen presenting cells (APCs). It is challenging to assess the potency of BsAb that engages two different signaling pathways simultaneously in a single assay format, especially when the two antigen targets are expressed on different cells. To explore the potency of anti-PD-L1/CD40L BsAbFP, a fusion protein that binds to human CD40 and PD-L1, we engineered CHO cells as surrogate APCs that express T cell receptor activator and PD-L1, Jurkat cells with PD-1 and NFAT-luciferase reporter as effector T cells, and Raji cell with NFkB-luciferase that endogenously expresses CD40 as accessory B cells. A novel reporter gene bioassay was developed using these cell lines that allows anti-PD-L1/CD40L BsAbFP to engages both PD-1/PD-L1 and CD40/CD40L signaling pathways in one assay. As both reporters use firefly luciferase, the effects of activating both signaling pathways is observed as an increase in luminescence, either as a higher upper asymptote, a lower EC_50_, or both. This dual target reporter gene bioassay system reflects potential mechanism of action and demonstrated the ability of anti-PD-L1/CD40L BsAbFP to synergistically induce biological response compared to the combination of anti-PD-L1 monovalent monoclonal antibody and agonist CD40L fusion protein, or either treatment alone. The results also showed a strong correlation between the drug dose and biological response within the tested potency range with good linearity, accuracy, precision, specificity and stability indicating properties, suggesting that this “three-cell-in-one” dual target reporter gene bioassay is suitable for assessing potency, structure-function and critical quality attributes of anti-PD-L1/CD40L BsAbFP. This approach could be used for developing dual target bioassays for other BsAbs and antibodies used for combination therapy.

## 1. Introduction

Immunotherapeutic antibodies have demonstrated superior tolerability and major improvements in long-term survival in cancer patients [1,2]. Use of immune checkpoint inhibitors or costimulatory molecules as a target for immunotherapy has become one of the promising approaches in the field of cancer therapy. Among antibody-based cancer therapies, bispecific antibodies (BsAbs), and antibody-fusion proteins (BsAbFPs), that bind to two different antigen targets, are an emerging class of biotherapeutics for cancer patients [3]. Following the regulatory approval of blinatumomab, a number of BsAbs have undergone clinical development. BsAbs can be classified as cytotoxic effector cell redirectors, tumor-targeted immunomodulators and dual immunomodulators. In recent years, numerous BsAbs targeting the costimulatory (e.g., OX40, CD40, 4-1BB etc.), coinhibitory pathways (PD-1, CTLA-4, TIM-3, TIGIT etc.) or tumor antigens (MUC1, MUC16) have been developed and are in preclinical and clinical investigations. The BsAb format is often a combination of two distinct variable regions derived from two different parental monospecific antibodies. In other cases, such as that presented here, both Fab arms bind the same target, and an additional target-specificity is added via protein fusion on the C-terminus of the Fc domain. Unlike binding to single-antigen on target cells, BsAbs possess the ability to bind to two antigens simultaneously. The ability of BsAbs to simultaneously bind two different antigens enables a variety of potential mechanism of actions (MOAs), such as an improved cytotoxic potential by bridging cells in-trans, synergistic effects, and receptor crosslinking leading to enhanced inhibition or degradation of target proteins from the cell surface with higher binding specificity and tumor selectivity [4].

Programmed death 1 (PD-1) is expressed on activated T cells and has emerged as an important mediator for negatively regulating T cell responses. It acts as a key checkpoint molecule in tumor-induced immune suppression [5]. Antibodies targeting PD-1/PD-L1 checkpoint stimulate the immune system to keep the tumor in check by promoting the T cells response from attenuating effects of PD-1. When engaged by one of its ligand PD-L1 or PD-L2, PD-1 inhibits the kinases that are involved in T cell activation and downstream signaling pathways that further suppress the effector function including exhaustion of T cell immune response [6,7]. Blockade of the interaction of PD-1 with PD-L1 or PD-L2 has been shown to enhance the antitumor activity of T cells [7,8]. 

CD40 is a membrane protein belonging to TNF receptor (TNFR) superfamily that functions as a key costimulatory molecule for activating both innate and adaptive immune system [9]. It also has a non-redundant role in antibody class-switching [10]. It is primarily expressed on the surface of B cells, dendritic cells, macrophages and on most of their neoplastic counterparts [11]. CD40 binding to its ligand (CD40L) exerts profound effects on these cells, and regulates host immune response against various pathogens [12]. Ligation of CD40 to CD40L results in cellular activation for generation of both T cell-dependent and humoral immune responses in cancer and autoimmune disorders [13]. CD40 signaling in B cells activates NF-κB and other cellular signaling pathways [14]. CD40 agonist antibodies have demonstrated anti-tumor activity but with dose-limiting toxicity [9,15]. They also promote APC maturation, license cross-presentation of antigens to cytotoxic T cells, enhance macrophage tumoricidal activity, and may therefore alter the tumor microenvironment to increase sensitivity to immune checkpoint blockade [16,17].

Although immunomodulators-based monotherapy targeting the costimulatory or coinhibitory pathways have demonstrated profound clinical efficacy for multiple cancer indications, clinical applications of these therapies are restricted due to limited therapeutic effects and survival benefits found only in a subset of patients [7,18]. For example, combined dual costimulation through 4-1BB and OX40 induced profound CD8^+^ T clonal expansion and synergistic responses [19]. Antitumor effects of combined anti-PD-1 mAb and anti-OX40 mAb in a murine ovarian cancer model increased the activity of effector CD4^+^ and CD8^+^ T cells and attenuated the action of immunosuppressive CD4^+^FoxP3^+^ regulatory T cells in tumor sites [20]. A study by Singh et al. [21] showed that intra-tumoral CD40 activation and checkpoint blockade induced T cell-mediated eradication of melanoma in the brain. In this study, through engaging the two different antigen targets (PD-L1, CD40) expressing cells together with PD-1 expressing T cells, we demonstrated that anti-PD-L1/CD40L BsAbFP simultaneously activates the costimulatory CD40 pathway in CD40-expressing B cell and inhibits the coinhibitory PD-1 pathways in T cell, leading to synergistic effect of the BsAbFP which was not achieved by combination of two mono-target molecules using anti-PD-L1 mAb and CD40 ligand. Our results also showed a strong correlation between the drug dose and biological response within the tested potency range and the dual target reporter bioassay is target specific and stability indicating.

## 2. Results

### 2.1. Generation of Anti-PD-L1/CD40L Bispecific Fusion Protein

A full length IgG4 anti-PD-L1/CD40L BsAbFP was generated within AstraZeneca. Anti-PD-L1/CD40L BsAbFP contains the N-terminal anti-PD-L1 arm composed of anti-PD-L1 mAb, and the CD40 agonist CD40L tripeptide arm at the C-terminal of the Fc region connected through amino acid linkers (Figure 1a). The CD40L tripeptide folds into a trimer confirmation closely resembling that of the native CD40L trimer [22]. The anti-PD-L1/CD40L BsAbFP was purified to homogeneity by protein A column and subjected to purity analysis by size-exclusion chromatography and mass spectroscopy. The analytical size-exclusion chromatography showed anti-PD-L1/CD40L BsAbFP purity of 97.7% with low levels of high molecular weight aggregates (1.5%) and fragment (0.8%). The high product quality of purified anti-PD-L1/CD40L BsAbFP was confirmed by mass spectrometry (MS) analysis and no heterogenous antibody by-products such as fragments or half antibodies lacking CD40L were detectable (data not shown). Based on the structure of the molecule, anti-PD-L1/CD40L BsAbFP had a potential MOA that activated costimulatory CD40 signaling in B cells via CD40L and inhibited the coinhibitory PD-1/PD-L1 pathway via anti-PD-L1 arm leading to T cell activation (Figure 1b).

### 2.2. Assessment of Anti-PD-L1 Activity of Anti-PD-L1/CD40L BsAbFP

Efficient T cell function regulation depends on co-clustering of T cell receptor (TCR) and other co-signaling complexes at the immune synapse. Binding of PD-L1 expressed on the APCs to PD-1 expressed on the activated T cell results in anergy and apoptosis of functional T cells [23]. Blockage of coinhibitory molecules PD-1/PD-L1 can enhance T cell function promoting proliferation, cytokine production and cytotoxicity [24]. To confirm the biological activity of anti-PD-L1 arm of anti-PD-L1/CD40L BsAbFP, we developed a PD-L1 bioassay using Jurkat T cells engineered to express PD-1/NFAT-Luciferase (Jurkat/PD-1/NFAT-Luc) and CHO cells engineered to express TCRA and PD-L1 (Figure 2a). The cells were treated with serial dilution of anti-PD-L1/CD40L BsAbFP or mono-target anti-PD-L1 mAb. As shown in Figure 2c, the anti-PD-L1/CD40L BsAbFP and the anti-PD-L1 mAb each activated Jurkat/PD-1/NFAT-Luc cell in a dose-dependent manner. Further, the dose-response curves are extremely similar, with upper and lower asymptotes and slopes sufficiently close to allow parallelism comparisons. The EC_50_ values of anti-PD-L1/CD40L BsAbFP and anti-PD-L1 mAb were 36.80 ± 3.37 ng/mL (mean ± SD) and 43.99 ± 2.88 ng/mL, respectively. These results indicated that the anti-PD-L1 arm of anti-PD-L1/CD40L BsAbFP possessed anti-PD-L1 activity leading to T cell activation, equivalent to anti-PD-L1 mAb. 

### 2.3. Assessment of CD40L Activity of Anti-PD-L1/CD40L BsAbFP 

CD40 signals B cell survival in part via transcriptional activation of the NF-κB pathway [25]. In order to assess the ability of CD40L arm in anti-PD-L1/CD40L BsAbFP to stimulate CD40 signaling pathway, we developed a CD40L bioassay using Raji cells expressing NFκB-Luciferase (Raji/NFκB-Luc) (Figure 2b). The cells were treated with a serial dilution of anti-PD-L1/CD40L BsAbFP or CD40L fusion protein. As shown in Figure 2d, anti-PD-L1/CD40L BsAbFP and CD40L fusion protein stimulated target cells (Raji/NFκB-Luc) in a dose-dependent manner with EC_50_ of 6.62 ± 2.99 ng/mL for CD40L and 10.11 ± 0.63 ng/mL for anti-PD-L1/CD40L BsAbFP, respectively. As shown above, curves of CD40L and BsAbFP were highly similar, including asymptotes and slopes sufficiently close to allow parallelism comparisons. These results confirmed that CD40 arm of anti-PD-L1/CD40LFP BsAb could activate the CD40 signaling pathway to elicit biological response in B cell. 

### 2.4. Design of “Three-Cell-in-One” Dual Target Reporter Gene Bioassay for Anti-PD-L1/CD40L BsAbFP 

The development of MOA-reflective bioassay to monitor the potency of BsAb is always challenging. The successful development of potency assay to monitor overall activities of anti-PD-L1/CD40L BsAbFP in a single assay format is even more challenging because the two target antigens (PD-L1, CD40) are not biologically expressed on the same target cell, or pair of target cells. We designed a novel in vitro bioassay with “three-cell-in-one” format that engaged three cell lines together in the same well that imitated the potential MOA of anti-PD-L1/CD40L BsAbFP. As shown in Figure 3a, the assay engaged the anti-PD-L1/CD40L BsAbFP to its respective target antigens (PD-L1, CD40) in engineered CHO/PD-L1/TCRA cells and Raji/NFkB-Luc cells. Combining these cells together with Jurkat/PD-1/NFAT-Luc cells yielded an assay which measured T cell and APC activation. In the presence of target cells expressing PD-L1, anti-PD-L1 arm of anti-PD-L1/CD40L BsAbFP disrupted the PD-L1/PD-1 coinhibitory pathway leading to Jurkat T cell activation. In the presence of Raji/NFκB-Luc cells, binding of CD40L arm of anti-PD-L1/CD40L BsAbFP to CD40 led to Raji/NFkB-Luc cell activation. Both the Jurkat and Raji cells utilized firefly luciferase as a reporter, so the combined expression was inferred from the additive or synergistic increases in luciferase activity observed when both ligands were targeted, as compared to either one alone.

### 2.5. Potency Assessment of Anti-PD-L1/CD40L BsAbFP by Dual Target Reporter Gene Bioassay 

Based on the assay design illustrated in Figure 3a, all three cell lines were co-cultured for 4 hr in presence of (1) anti-PD-L1/CD40L BsAbFP, (2) anti-PD-L1 mAb plus CD40L fusion protein, (3) anti-PD-L1 mAb alone or (4) CD40L fusion protein alone. As shown in Figure 3b, moderate drug response was seen in the cells treated with either anti-PD-L1 mAb or CD40L fusion protein alone, indicating that the assay is responsive to both antigen targets (PD-L1, CD40). Compared to cells treated with either anti-PD-L1 mAb or CD40L fusion protein alone, the combination treatment of anti-PD-L1 and CD40L fusion protein showed substantially increased (additive) response in a concentration-dependent manner, specifically, an increase in the upper asymptote. Furthermore, compared to other groups, treatment with anti-PD-L1/CD40L BsAbFP showed the strongest drug response that was even higher than the combination treatment with anti-PD-L1 mAb plus CD40L fusion protein, including both a further increase in the upper asymptote and a left-ward shift in the EC_50_. (Figure 3b). The EC_50_ values were 2.87 ng/mL (BsAbFP), 52.96 ng/mL (anti-PD-L1 mAb), 3.91 ng/mL (CD40L) and 24.99 ng/mL (anti-PD-L1 mAb + CD40L). These results indicate that the anti-PD-L1/CD40L BsAbFP enabled a synergistic effect compared to combination treatment. The possible reasons for this are addressed in the Discussion. Overall, these data demonstrated that via simultaneous engagement of anti-PD-L1/CD40L BsAbFP with its two target antigens-expressing cells, the assay was capable of capturing the biological activity of anti-PD-L1/CD40L BsAbFP which was significantly higher than either treatment alone, or the combination of the two monospecific molecules. 

### 2.6. Characterization of PD-L1/CD40 Dual Target Reporter Bioassay

#### 2.6.1. Assessment of the Role of Raji/NFkB-Luc Cells on the PD-L1/CD40L Dual Target Reporter Bioassay

To assess the role of Raji/NFκB-Luc cell on the PD-L1/CD40L dual target reporter bioassay, the Raji/NFκB-Luc cells were replaced by unmodified Raji cells (Raji/NL) that did not contain an NFκB-luciferase reporter gene, but still expressed CD40 (Figure 4a). In this assay using CHO/PD-L1/TCRA cell + Raji/NL cell + Jurkat/PD-1/NFAT-Luc cell, the response signal of anti-PD-L1/CD40L BsAbFP was weaker compared to the normal PD-L1/CD40L dual target bioassay (CHO/PD-L1/TCRA cell + Raji/NFκB-Luc cell + Jurkat/PD-1/NFAT-Luc cell) (Figure 4d). The results confirmed that Raji/NFκB-Luc cells played an important role on the PD-L1/CD40L dual target reporter bioassay.

#### 2.6.2. Assessment of the Role of Jurkat/PD-1/NFAT-Luc Cells on the PD-L1/CD40L Dual Target Reporter Bioassay

To determine the role of Jurkat/PD-1/NFAT-Luc cells on the PD-L1/CD40L dual target reporter bioassay, the Jurkat/PD-1/NFAT-Luc cells were replaced by unmodified Jurkat (Jurkat/NL) cells that did not express PD-1 and a NFAT-luciferase report gene (Figure 4b). In this modified assay (CHO/PD-L1/TCRA cell + Raji/NFκB-Luc cell + Jurkat/NL cell), the signal of anti-PD-L1/CD40L BsAbFP was lower compared to the normal PD-L1/CD40L dual target bioassay (Figure 4d). These results confirmed that Jurkat/PD-1/NFAT-Luc cell played an important role on the PD-L1/CD40L dual target reporter bioassay. Furthermore, if the assay system only contained CHO/PD-L1/TCRA cells and unmodified Raji cells but not Jurkat/PD-1/NFAT-Luc (Figure 4c), as expected, no response signal was observed for anti-PD-L1/CD40L BsAbFP (Figure 4d). 

### 2.7. Stability and Specificity Assessment of PD-L1/CD40 Dual Target Reporter Bioassay 

Next, we determined if the PD-L1/CD40L dual target reporter gene bioassay was specific to the target antigens, PD-L1 and CD40. The specificity of the assay was assessed by incubating the cells with irrelevant antibody (mAb-X, a human IgG1 mAb) or formulation buffer. As expected, low basal levels of luciferase reporter gene were detected with the non-specific mAb or buffer alone (Figure 5a). Only the anti-PD-L1/CD40L BsAbFP treatment showed dose-dependent response indicating that the activation of costimulatory CD40 and inhibition of coinhibitory PD-1 signaling pathways were target-specific. We also sought to determine if the “three-cell-in-one” dual target reporter gene assay was sensitive enough to detect the activity of forced degraded anti-PD-L1/CD40L BsAbFP test samples. We incubated the cells with heat-stressed (40 °C for 7 weeks) or cool white light + UV treated (CWL + UV) anti-PD-L1/CD40L BsAbFP and measured the potency. As expected, compared to control group, potencies of the thermal and photo-stressed anti-PD-L1/CD40L BsAbFP decreased by 28% and 64%, respectively, indicating that the dual target bioassay was able to detect the potency change of anti-PD-L1/CD40L BsAbFP due to thermal or photo (CWL + UV) stress (Figure 5b). 

### 2.8. Linearity Assessment of PD-L1/CD40 Dual Target Reporter Assay

We next assessed the relationship between the anti-PD-L1/CD40L BsAbFP concentration and signaling response by analyzing the ability of anti-PD-L1/CD40L BsAbFP to activate the respective signaling pathways and response as luciferase gene expression within a linear range of 60–167% relative potency (RP). Parallelism between reference standard and test samples was assessed in all assays using system suitability criteria for each assay and sample, including comparison of upper and lower asymptotes and slope. As shown in Figure 6a, anti-PD-L1/CD40L BsAbFP activated reporter gene expression in a dose-dependent manner. A prepared 60% RP test sample showed lower activity than the reference standard (curve shifts to the right) while a prepared 167%RP sample showed stronger activity than the reference standard (curve shifted to the left). We plotted the dose-response results in the regression model to determine the linearity of the responses (Figure 6b). The linear regression plot showed R^2^ = 0.996, indicating a strong correlation between the drug dose and biological response within the tested potency range of 60–167% RP. 

### 2.9. Qualification of PD-L1/CD40L Dual Target Reporter Bioassay

The qualification of the PD-L1/CD40L dual target reporter bioassay was performed according to ICH Q2(R1) guideline. The assay qualification results are summarized in Table 1. In addition to good linearity (R^2^ = 0.996), the assay’s repeatability (*n* = 6) and intermediate precision (*n* = 6) were 7.7% and 14.2%, respectively. The accuracies of the assay across five potency levels (60%, 77%, 100%, 130% and 167%) were 92% to 109.8% (Table 1). Together with the specificity and stability indicating assessment described in the section above, we successfully demonstrated that the PD-L1/CD40L dual target reporter bioassay was a qualifiable assay that could be used in GMP lot release and stability testing for product potency determination.

### 2.10. Evaluation of Structure-Function Attributes Using Different Bioassays and Analytical Methods

A critical product quality attribute has the potential to affect the safety and efficacy of therapeutic proteins. Bioassays play an important role on assessing the structure-function attributes during product development, product quality control and evaluation on the impact of a critical quality attributes (CQA). To assess the impact of modification of complementarity determining region (CDR) (anti-PD-L1) or ligand binding site (CD40L) due to high pH stress (pH = 8.5) on anti-PD-L1/CD40L BsAbFP and its criticality of the quality attributes, we evaluated the biological activity using three different bioassays, CD40L bioassay, PD-L1 bioassay, and PD-L1/CD40L dual target reporter bioassay. Compared to the control group, CD40L bioassay results indicated that the percent relative potency (%RP) of CD40L arm was greatly decreased at 2 wk (55.2%) and 4 wk (30.2%), suggesting that the CD40L arm of anti-PD-L1/CD40L BsAbFP that contains a CD40L trimer was strongly impacted under these stress conditions. The %RP of anti-PD-L1 arm, on the other hand, was relatively stable under high pH stress condition at 2 wk (78.9%) and 4 wk (66.0%) as determined using the PD-L1 bioassay. The results from PD-L1/CD40L dual target reporter bioassay also showed time-dependent activity decrease at 2 wk (72.5%) and 4 wk (58.9%), respectively (Figure 7a). The size exclusion chromatography (SEC) results showed an increase in aggregation and CD40L clipping peaks over time (Figure 7b), which supported our finding of reduced potency using the CD40L bioassay. A further evaluation of the increased peaks for major post-translational modification (PTM) by the peptide mapping results revealed that high pH gradually induced the oxidation in the anti-PD-L1 heavy chain and dramatically increased the deamidation in the CD40L arm (Figure 7c). These results from SEC and peptide mappings aligned and supported our finding for potency results due to impact of each arm as assessed by three different bioassays. Taken together, these results revealed that PD-L1/CD40L dual target reporter bioassay measures the potency of anti-PD-L1/CD40L BsAbFP as a whole, while two single target bioassays (CD40L bioassay, PD-L1 bioassay) could be used to characterize the biological activity of each arm of anti-PD-L1/CD40L BsAbFP and CQA assessment, to demonstrate the relative contribution of each arm to stability.

## 3. Discussion

Bispecific antibodies and fusion proteins have numerous potential clinical applications including redirection of cellular activity by recruitment of different cell types to each other or modification of cell signaling pathways. They have the potential to improve clinical efficacy and safety. A wide range of preclinical and clinical data indicate important roles of coinhibitory and costimulatory signals in the progression of cancer or infectious diseases and provide a strong rationale for the combined targeting of these signaling molecules [26]. Several families of inhibitory receptors and stimulatory receptors have been found to be important in regulating T cell responses. Combined inhibition or activation of these receptors may contribute to T cell activation and function. One of the advantages of BsAbs is that they enable unique MOAs that may be not achievable with monospecific or combination therapies. A recent publication from Chen et al., for example, showed synergistic effect of anti-PD-1/CTLA4 BsAb targeting human CTLA-4 and PD-1 in CTLA4/PD-1-expressing target cells [27]. 

Activation of tumor-specific T cell responses has been shown to be strongly affected by activation of the CD40 signaling pathways on the APCs [28]. We have constructed a novel bispecific fusion protein, anti-PD-L1/CD40L BsAbFP, based on the strong scientific rationale that coupling of CD40 stimulation signaling together with inhibition of PD-1 coinhibitory pathway that rescues T cells from exhaustion, enhances T cell priming and tumor-infiltrating lymphocytes (TIL) activation in mouse models of nonimmunogenic solid malignancies [29,30]. The biological function of anti-PD-L1 mAb and CD40L was studied in the B16F10 syngeneic model where anti-PD-L1/CD40L BsAbFP effectively inhibited tumor growth and induced robust INF-γ production [31]. Furthermore, the study demonstrated that anti-PD-L1/CD40L BsAbFP drives the internalization and degradation of PD-L1 by bringing two target molecules together, while simultaneously stimulating the CD40 pathway and inhibiting the coinhibitory PD-1 pathway. 

As potency is one of the major CQAs, specifically a default CQA for a biologic, accurately measuring potency using appropriate bioassay plays an important role during manufacturing process optimization and product quality control. Indeed, per regulatory agency guideline, the potency bioassay needs to reflect the potential MOA of the drug as much as possible. Using single target bioassays, PD-L1 bioassay and CD40 bioassay, the results of a characterization study of anti-PD-L1/CD40L BsAbFP confirmed the ability of each arm to interact with PD-L1 and CD40 in ways that are similar to the parental anti-PD-L1 mAb or CD40L fusion protein. For BsAb, however, it is very challenging to measure the potency of the BsAb by assessing the two different signaling pathways simultaneously in a single assay format. The complexity further increases when the two different antigen targets are expressed on different cells and perhaps a key signal emanates from a third. We developed and established a novel method permitting selective binding of CD40L and anti-PD-L1 mAb in a single assay format where two different target antigens (PD-L1 and CD40) were expressed on different cell lines, and interact with a third cell. This assay was able to simultaneously detect binding to both antigens and activation of two different signaling pathways, thus mimicking the potential MOA and putative therapeutic activity of the anti-PD-L1/CD40L BsAbFP. A recent study using anti-CD40 mAb and anti-PD-1/CTLA4 BsAb showed the therapeutic benefit over combination therapy of mAbs by recapitulating the in vivo synergistic effect [32]. These results indicate that activation of two independent T cell signaling pathways could synergistically benefit towards combination therapy.

Our results imply that targeting PD-1 and CD40 signaling pathways using combination treatment only induced an additive effect, whereas anti-PD-1/CD40L BsAbFP induced a synergistic effect. The molecular basis for anti-PD-L1/CD40L BsAbFP induced synergistic response is unknown but the following possibilities may contribute: (i) by simultaneous binding of PD-L1 and CD40, the dual-antigen binding by anti-PD-1/CD40L BsAbFP stabilizes the interaction between PD-L1 and anti-PD-L1 mAb, and CD40 and CD40L, that may enhance the downstream activation of signaling pathways (Figure 3); (ii) the engagement of anti-PD-L1/CD40L BsAbFP to the two target antigens may enhance the internalization and lysosomal degradation of the target protein-antibody complexes, that results in sustained activation of costimulatory and inhibition of coinhibitory signals; (iii) engagement of the two target antigens with anti-PD-L1/CD40L BsAbFP brings three target cells in close proximity that enhances the cross-talk between the B and T cells for other cellular events. In contrast, combination of two single-target mAbs or mAb alone lack all those above mentioned characteristics that results in a reduced/moderate level of independent signaling events. Although the third possibility is technically difficult to test in the current setting, our results support the reports published by Yaya et al. (2020) [31]. Furthermore, our preliminary results showed that treatment with anti-PD-L1/CD40L BsAbFP increased CD69 expression (early T cell activation marker) on the surface of the activated T cells compared to the combination or mAb alone (data not shown). 

Through substituting the Raji/NFκB-Luc cell with unmodified Raji cell or replacing Jurkat/PD-1/NFAT-Luc cell with unmodified Jurkat cell, the assay characterization study results confirmed that both Raji/NFκB-Luc cell and Jurkat/PD-1/NFAT-Luc cell play an important role on the dual target reporter bioassay, specifically generating enhanced luciferase reporter signaling upon anti-PD-L1/CD40L BsAbFP costimulation. Our results indicate that both anti-PD-L1 arm and CD40L arm of the BsAbFP were engaged in this dual target bioassay. The assay qualification results demonstrated that the PD-L1/CD40L dual target reporter bioassay has good accuracy, precision, specificity and stability indicating property, suggesting the assay is suitable for measuring the potency of anti-PD-L1/CD40L BsAbFP in quality control system for lot release and stability. Further studies with different ratio of Jurkat/PD-1/NFAT-Luc cell: CHO/PD-L1/TCRA cell: Raji/NFκB-Luc cell could be performed during assay robustness assessment to further optimize the assay condition. Additionally, as the signal duration is different between PD-1 and CD40 signaling activation, it is important to identify an appropriate incubation time range that works for both signaling pathways when assessing different cell ratios.

As bioassays play an important role on assessing structure-function attributes, potential CQAs and understanding of the molecule, we characterized the impact of high pH stress on anti-PD-L1/CD40L BsAbFP’s biological activities using the CD40L bioassay, PD-L1 bioassay and CD40L/PD-L1 dual target reporter bioassay. Although the two single-target bioassays (CD40L bioassay, PD-L1 bioassay) were suitable for measuring the activity of each arm of the molecule, none of these two measurements represents the overall potency of anti-PD-L1/CD40L BsAbFP. In other words, among these three bioassays, only PD-L1/CD40L dual target reporter bioassay is a suitable assay for measuring the potency of the intact anti-PD-L1/CD40L BsAbFP as a whole. As the secondary and tertiary structures of CD40L arm and anti-PD-L1 arm are very different, it is not unusual to see different impact on each arm under high pH stress. To specifically assess the impact of pH stress on each arm of the molecule, CD40L bioassay and PD-L1 bioassay can be good tools for this purpose. The results from the SEC and peptide mapping are well aligned with the results obtained from CD40L bioassay and PD-L1 bioassay, suggesting these single-target assays are useful bioassays for assessing the structure-function attributes of each arm of anti-PD-L1/CD40L BsAbFP since the dual target bioassay cannot specifically distinguish the two arms’ potencies of the molecule. 

In summary, our results show that anti-PD-L1/CD40L BsAbFP simultaneously interacts with the target antigens and stimulates costimulatory CD40 pathway and inhibits coinhibitory PD-1/PD-L1 pathway exhibiting a synergistic effect of the BsAbFP. We further confirmed that the synergistic response is only achieved by anti-PD-L1/CD40L BsAbFP but not combination of mAbs or either treatment alone. Together with two single target bioassays (PD-L1 bioassay, CD40L bioassay), the dual target bioassay could be useful for the CQA and potency assessment of anti-PD-L1/CD40L BsAbFP. As the dual target potency bioassay is more MOA-reflective than single target bioassays, this approach could be used for assessing the biological activity of other BsAbs and antibodies used for combination therapy. 

## 4. Materials and Methods

### 4.1. Cell Lines and Reagents

Jurkat cell line engineered to express PD-1 and NFAT-Luciferase (Jurkat PD-1/NFAT/Luc) and Steady-Glo luciferase reagent were purchased from Promega (Madison, WI, USA). CHO cell line was engineered in-house to express T cell receptor activator (TCRA) and PD-L1 (CHO/PD-L1/TCRA) also referred to as APC. Raji cells endogenously expressing CD40 was engineered in-house to express NFkB/Luciferase (Raji/NFkB-Luc). All cell lines were routinely cultured in their respective cell culture medium. Assay ready cell (ARC) banks were generated and used in the experiments. All cell lines were tested for the absence of mycoplasma. APC culture medium contains DMEM (ThermoFisher Scientific, Waltham, MA, USA, Cat# 11995) supplemented with 10% heat inactivated (HI) FBS. Assay medium contains RPMI 1640 GlutaMAX, HEPES (ThermoFisher Scientific, Cat# 72400) supplemented with 10% HI FBS.

### 4.2. PD-1 Bioassay

Frozen vials of CHO/PD-L1/TCRA cells and Jurkat/PD-1/NFAT-Luc cells were prepared in CHO/PD-L1/TCR cell culture and assay medium, respectively. CHO/PD-L1/TCRA cell at 1 × 10^4^ cells/well and Jurkat/PD-1/NFAT-Luc cells at 2 × 10^4^ cells/well were co-cultured in 96-well flat bottom plates for 16–22 h at 37 °C, 5% CO_2_ in the presence of anti-PD-L1/CD40L BsAbFP or anti-PD-L1 mAb. The initial antibodies concentration were set at 3000 ng/mL with a 2.6-fold serial dilution. The next day, luciferase reporter gene expression was determined by measuring the amount of luminescence using Steady-Glo luciferase assay system (Promega) and Envision plate reader (PerkinElmer, Waltham, MA, USA) following manufacturers’ instruction. Three independent experiments in triplicate were performed to measure the activity of anti-PD-L1/CD40L BsAbFP and anti-PD-L1 mAb. 

### 4.3. CD40 Bioassay

Frozen vials of Raji/NFkB-Luc cell were prepared in assay medium. Cells at 5 × 10^5^ cells/well were incubated in 96-well flat bottom plates for 4 h at 37 °C, 5% CO_2_ in the presence of anti-PD-L1/CD40L BsAbFP or CD40L fusion protein. The initial concentration was set at 1000 ng/mL with a 3.0-fold serial dilution. After indicated time, cells were processed for luciferase reporter gene expression by measuring the amount of luminescence following manufacture instruction (Promega). Three independent experiments in triplicate were performed to measure the activity of anti-PD-L1/CD40L BsAbFP or CD40L fusion protein. 

### 4.4. PD-L1/CD40L Dual-Target Reporter Gene Bioassay

Frozen vials of CHO/PD-L1/TCRA cell were thawed, prepared in CHO/PD-L1/TCRA cell culture medium and 100 µL per well cell suspension at 5.0 × 10^5^ cells/mL (5.0 × 10^4^ cells/well) were seeded into the center sixty wells of a white 96-well tissue culture plate. Next day, a serial dilution of anti-PD-L1/CD40L BsAbFP at a starting concentration of 400–3300 ng/mL was prepared in assay medium. The CHO/PD-L1/TCRA cell medium was discarded from the assay plates and a fresh 40 µL per well assay medium was added into each well. A 40 µL/well of anti-PD-L1/CD40L BsAbFP, prepared as above, was added into the wells. During sample pre-incubation, frozen aliquots of Jurkat/PD-1/NFAT-Luc cells and Raji/NFκB-Luc cells were thawed, processed and seeded at 1.0 × 10^6^ cells/mL and 2.0 × 10^6^ cells/mL, respectively. The two cell solutions were mixed and 40 µL cell mixture per well was added into the assay plates. The plates were incubated at 37 °C, 5% CO_2_ incubator for 4.5 ± 0.5 h. After the indicated incubation time, 100 μL per well Steady-Glo^TM^ reagent was added to each well, and plates were shaken for 30–45 min at room temperature before measuring luminescence using an Envision plate reader. The luminescence results were plotted as RLU (relative luminescence units) versus concentration of anti-PD-L1/CD40L BsAbFP using 4PL parallel-line analysis to determine EC_50_ and potency as described below. Three independent experiments were performed in triplicate to measure the activity of anti-PD-L1/CD40L BsAbFP. 

### 4.5. Size Exclusion Chromatography (SEC)

The SEC method was conducted using a TSKgel G3000SWXL (7.8 × 300 mm, 5 µm; Tosoh Bioscience, San Francisco, CA, USA) and isocratic mobile phase of 0.1 M sodium phosphate and 0.1 M sodium sulfate, pH 6.8 running at 1.0 mL/min.

### 4.6. Peptide Mapping for PTM Measurements

The high pH-stressed protein at different time points were digested by trypsin. Digests of the sample were analyzed using a Fusion Orbitrap mass spectrometer (Thermo Fisher Scientific, Waltham, MA, USA) connected with an AQUITY ultra-performance liquid chromatograph (UPLC; Waters, Milford, MA, USA). An AQUITY UPLC BEH300 C18 column (1.7 μm, 2.1 × 150 mm; Waters) was used for separation. The column temperature was held at 55 °C. Mobile phase A was 0.02% TFA in water, and mobile phase B was 0.02% TFA in acetonitrile. Digested peptides were eluted from the column with a 0–35% linear gradient and the chromatographic profile was monitored by using UV absorbance at 220 nm and MS. MS data were processed by Biopharma Finder 3.0 (Thermo Fisher Scientific, Waltham, MA, USA).

## 5. Data Analysis

The dose-response curve was generated using a four parameter semi-logistical curve model through SoftMax Pro software (Molecular Device, San Jose, CA, USA). EC_50_ values represent the concentration of antibody at which half-maximal activation was observed. After assessing parallelism between reference standard and test samples, the percent relative potencies of the test samples were assessed by dividing the EC_50_ value of the reference standard by the EC_50_ value of each sample and multiplying by 100. The statistical analyses were performed using GraphPad Prism Version 8.0 (San Diego, CA, USA).

## Figures and Tables

**Figure 1 ijms-22-11302-f001:**
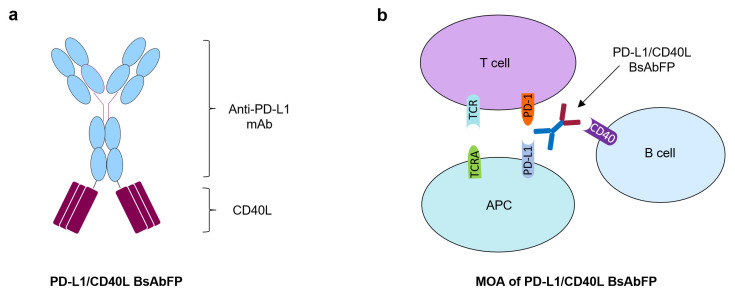
Structure illustration of anti-PD-L1/CD40L BsAbFP and potential mechanism of action. (**a**) Schematic representation of anti-PD-L1/CD40L BsAbFP consisting of anti-PD-L1 mAb and CD40L; (**b**) illustration of mechanism of action (MOA) of anti-PD-L1/CD40L BsAbFP that each arm binds to target antigens expressed on different cells, such as CD40-expressing B cells and PD-L1-expressing antigen presenting cells (APCs).

**Figure 2 ijms-22-11302-f002:**
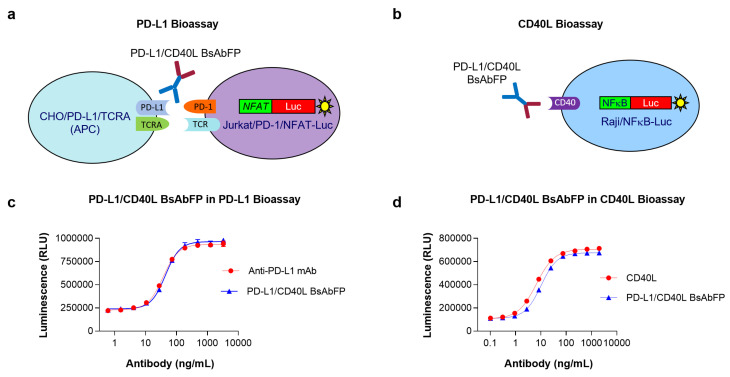
Assessment of each arm’s activity of anti-PD-L1/CD40L BsAbFP. (**a**) Illustration of PD-L1 bioassay: Anti-PD-L1 arm of anti-PD-L1/CD40L BsAbFP inhibits the PD-1/PD-L1 pathway leading to the activation of luciferase reporter gene in Jurkat/PD-1/NFAT-Luc cell; (**b**) Illustration of CD40L bioassay: CD40L arm of anti-PD-L1/CD40L BsAbFP binds to CD40 leading to the activation of luciferase reporter gene in Raji/NFκB-Luc cell; (**c**) Representative PD-L1 bioassay graph comparing dose-dependent response between of anti-PD-L1/CD40L BsAbFP and anti-PD-L1 mAb; (**d**) Representative CD40L bioassay graph comparing dose-dependent response between anti-PD-L1/CD40L BsAbFP and CD40L.

**Figure 3 ijms-22-11302-f003:**
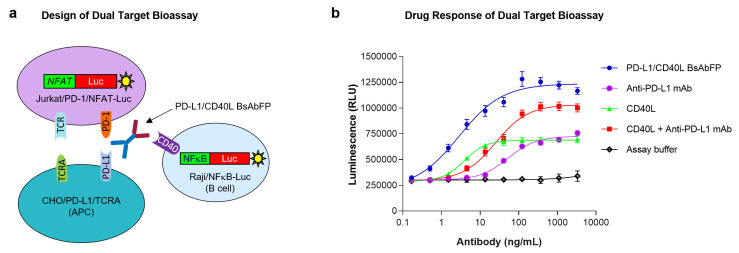
Development of dual target reporter gene bioassay for anti-PD-L1/CD40L BsAbFP. (**a**) Illustration of PD-L1/CD40L dual target reporter gene bioassay: binding of PD-L1 and CD40 targets by anti-PD-L1/CD40L BsAbFP forms a trimeric complex in the presence of Jurkat/PD-1/NFAT-Luc cell; (**b**) anti-PD-L1/CD40L BsAbFP, anti-PD-L1 mAb, CD40L fusion protein and combo (anti-PD-L1 mAb + CD40L) were tested in the dual target bioassay. The assay demonstrates that anti-PD-L1/CD40L BsAbFP strongly activates target cells compared to the combination treatment of the anti-PD-L1 and CD40L or either treatment alone. The assay was performed using a 1:3 serial dilution of antibody with a starting concentration of 3300 ng/mL.

**Figure 4 ijms-22-11302-f004:**
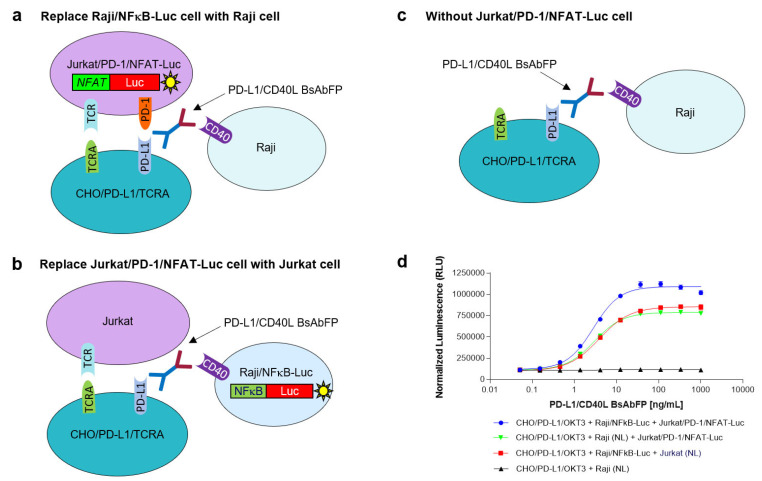
Characterization of PD-L1/CD40L dual target reporter bioassay. (**a**) Replacing Raji/NFκB-Luc cell with unmodified Raji cell (Raji/NL) that does not contain a luciferase reporter gene; (**b**) replacing Jurkat/PD-1/NFAT-Luc cell with unmodified Jurkat cell (Jurkat/NL) that does not contain a luciferase reporter or PD-1 gene; (**c**) use of unmodified Raji cell (Raji/NL) but without Jurkat/PD-1/NFAT-Luc cells; (**d**) summary of drug-response activity of anti-PD-L1/CD40L BsAbFP. The assay was performed using a 1:3 serial dilution of anti-PD-L1/CD40L BsAbFP with a starting concentration of 1000 ng/mL.

**Figure 5 ijms-22-11302-f005:**
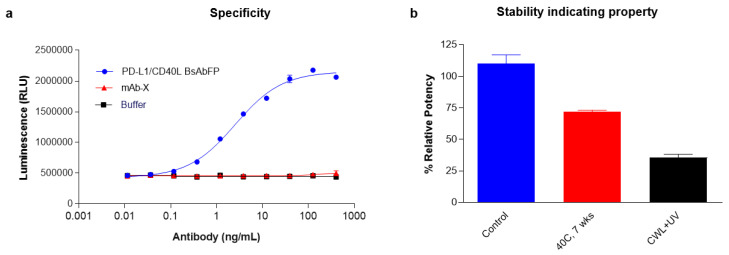
Specificity and stability indicating property assessment for the dual target reporter bioassay. (**a**) Specificity of the dual target reporter bioassay was assessed by testing the potency of anti-PD-L1/CD40L BsAbFP (blue), non-specific IgG (mAb-X, red) and formulation buffer (black). The dose-response data obtained were analyzed with a 4PL curve fit; (**b**) anti-PD-L1/CD40L BsAbFP samples were forced degraded with heat-stress (7 weeks at 40 °C) or UV light and assessed for potency using the dual target reporter bioassay. The results showed that dual target reporter bioassay is specific and can detect the changes in potency of forced degradation anti-PD-L1/CD40L BsAbFP samples. The assay was performed using a 1:3.2 serial dilution of anti-PD-L1/CD40L BsAbFP with a starting concentration of 400 ng/mL.

**Figure 6 ijms-22-11302-f006:**
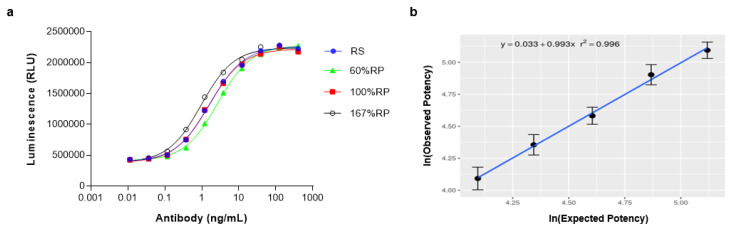
Linearity of dual target reporter gene bioassay for anti-PD-L1/CD40L BsAbFP. (**a**) Representative linearity curve fit at 60%, 100% and 167% relative potency of anti-PD-L1/CD40L BsAbFP; (**b**) a linear regression assessment for 60–167% relative potency samples from replicate runs (*n* = 6). Error bars indicate ± SD. The assay uses a 1:3.2 serial dilution of anti-PD-L1/CD40L BsAbFP with a starting concentration of 400 ng/mL.

**Figure 7 ijms-22-11302-f007:**
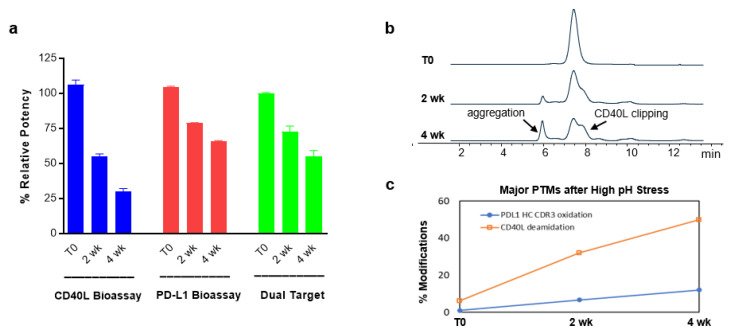
Evaluation of Structure-Function Attributes of Anti-PD-L1/CD40L BsAbFP. Potency attribute, molecular integrity, and post-translational modification (PTM) of anti-PD-L1/CD40L BsAbFP after high pH (pH = 8.5) stress were assessed using (**a**) CD40L bioassay, PD-L1 bioassay, and PD-L1/CD40L dual target reporter bioassay (Dual Target); (**b**) size exclusion chromatography (SEC) and (**c**) peptide mapping for PTM measurements of T0, 2 wk and 4 wk samples.

**Table 1 ijms-22-11302-t001:** Summary of assay qualification results.

Parameters	Acceptance Criteria	Results
Linearity Assay range	R^2^ ≥ 0.970 60–167% relative potency	R^2^ = 0.996 60–167% relative potency
Accuracy	80–125% across assay range 60–167% relative potency	Accuracy: 92.0–109.8%
Precision	Repeatability: %GCV: ≤15% Intermediate precision: %GCV: ≤20%	Repeatability: %GCV = 7.7% Intermediate Precision: %GCV = 14.2%
Specificity	No concentration-response to formulation buffer and structurally similar but functionally irrelevant antibody	No concentration-response for formulation buffer and irrelevant antibody
Stability Indicating	Detects a decrease in potency of stressed samples	Demonstrated a decrease in potency of UV- and thermal-stressed samples

## Data Availability

All published data used in the current study are available from the corresponding author on reasonable request.

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
