# Peer review of "Simultaneous Inhibition of PD-1 and Stimulation of CD40 Signaling Pathways by Anti-PD-L1/CD40L Bispecific Fusion Protein Synergistically Activate Target and Effector Cells"

_ijms, 2021, doi:10.3390/ijms222111302_

Round 1

Reviewer 1 Report

The authors reported that anti-PDL1/CD40L BsAbFP simultaneously interacts with the target antigens and stimulates the CD40 pathway and blocks the coinhibitory PD-1/PD-L1 pathway exerting a synergistic effect. But they did not clearly show the synergistic effect on target cells or antitumor efficacy of the BsAbFP. The authors also showed that the dual-targeting assays were useful to evaluate each arm’s activity of anti-PD-L1/CD40L BsAbFP. The anti-PD-L1/CD40L BsAbFP synergistically engaged both target cells (Jurkart/PD-1/NFAT-Luc cells and CD40+Raji/NFκB-Luc B cells) compared with a combination of anti-PDL1 mAb and CD40L, suggesting that the dual-target (PD-L1 and CD40L) bioassay was useful for the CQA and potency assessment of anti-PD-L1/CD40L BsAbFP and applicable for combination therapy.

  1.  What was the purity of this BsAbFP? Do you have HPLC data?
  2. For testing binding affinities to target, have you ever compared with the cell-based fluorescent method (X Yu et al.  J Immunol Methods 2017, 442:49-53)?
  3. In Figure 7. Evaluation of structure-function analyses, the relative potency of anti-PD-L1/CD40 BsAbFP decreased over time, particularly for CD40L bioassay. What are the reasons for the difference in stability between CD40L and anti-PD-L1? Does it attribute to the instability of the CD40L arm of the anti-PD-L1/CD40L BsAbFP?
  4. What about B cell activation after anti-PD-L1/CD40L BsAbFP treatment?
  5. If the anti-PD-L1/CD40L BsAbfP works, it should activate CD40(+) B cells and reinvigorate and activate exhausted T cells simultaneously. Did you analyze the function or activity of B cells and T cells? Did you have data of T cell activation or exhaustion markers including PD-1 and PD-L1 on T cells after treatment?
  6. In this manuscript, the authors did not enclose the anti-tumor efficacy of this fusion protein. If the in vivo data were included, it would be much informative.

Author Response

Re: Manuscript ID: ijms-1388346

Title: “Simultaneous inhibition of PD-1 and stimulation of CD40 signaling pathways by anti-PD-L1/CD40L bispecific fusion protein synergistically activate target and effector cells”

Authors: Pandey et al.

Dear Reviewer,

Thank you for giving us the opportunity to submit a revised manuscript for publication in the Journal.  We appreciate the time and effort that you dedicated to providing helpful suggestions on our manuscript. We have addressed all the comments from you and updated the manuscript accordingly. Please see below for a point-by-point response to your comments.

  1. What was the purity of this BsAbFP? Do you have HPLC data?

The purity of this BsAbFP is determined to be 97.7% by size exclusion chromatography with 1.5% aggregates and 0.8% fragments. We have updated Section 2.1 to include this information. We do not have HPLC data.

  1. For testing binding affinities to target, have you ever compared with the cell-based fluorescent method (X Yu et al.  J Immunol Methods 2017, 442:49-53)?

The BsAbFP material used in this study was generated for phase I clinical study. The target antigen binding of the molecule had been verified during drug lead identification and optimization. We have not compared the binding affinity with the cell-based fluorescent method published by X Yu et al in 2017. However, the ability of BsAbFP to interact with its target antigens, PD-L1 and CD40, was confirmed using PD-L1 bioassay (Fig 2c) and CD40L bioassay (Fig 2d).

  1. In Figure 7. Evaluation of structure-function analyses, the relative potency of anti-PD-L1/CD40 BsAbFP decreased over time, particularly for CD40L bioassay. What are the reasons for the difference in stability between CD40L and anti-PD-L1? Does it attribute to the instability of the CD40L arm of the anti-PD-L1/CD40L BsAbFP?

Yes. The difference in stability between CD40L and anti-PD-L1 is due to the instability of the CD40L arm. The CD40L and PD-L1 binding domains are located at two ends of the BsAbFP (Fig 1a). The stability of each domain and activity towards individual targets are independent of each other.  As discussed in section 2.10, CD40L undergoes deamidation and clipping under stressed conditions (Fig 7b & c), which lead to more dramatic loss of potency by the CD40L bioassay. In comparison, the PD-L1 domain is structurally more stable and less susceptible to degradation and potency loss.   

  1. What about B cell activation after anti-PD-L1/CD40L BsAbFP treatment?

As Raji cells are a human B cell line that is used in both CD40 bioassay and the dual target bioassay, and the results from these assays show strong cellular response to this BsAbFP (Fig 2d, Fig 3b, Fig 4b &d), we believe that anti-PD-L1/CD40L BsAbFP treatment will activate human B cells.   

  1. If the anti-PD-L1/CD40L BsAbFP works, it should activate CD40(+) B cells and reinvigorate and activate exhausted T cells simultaneously. Did you analyze the function or activity of B cells and T cells? Did you have data of T cell activation or exhaustion markers including PD-1 and PD-L1 on T cells after treatment?

No, we have not fully analyzed the function or activity of B cells and T cells. As indicated in Discussion section (Page 10), our preliminary results showed that treatment with anti-PD-L1/CD40L BsAbFP increased CD69 expression (early T cell activation marker) on the surface of the activated T cells compared to the combination or mAb alone. Analyzing the function or activity of B cells and T cells would be an interesting topic for future study.

  1. In this manuscript, the authors did not enclose the anti-tumor efficacy of this fusion protein. If the in vivo data were included, it would be much informative.

Thank you for the comment. There is no in vivo anti-tumor efficacy data available at this time. This would be valuable for a future research and publication.

These changes have addressed all of the comments from you. We appreciate your helpful suggestions that were very useful for revising this manuscript. I hope you now find that this manuscript is suitable for publication.

Sincerely,

Shihua Lin, PhD

Analytical Sciences - Bioassay Development

AstraZeneca

Reviewer 2 Report

Summary of the main findings of the study

The study presented by Pandey et al concerns the development of an in vitro potency bioassay for the characterization in GMP context of new molecules in the format of an anti-PD-L1 fusion with a trimeric mime of CD40L. This potency assay is totally innovative because it promises to test in a single well the activation of 2 signaling pathways, that of PD-1 on Jurkat NFAT-reporter cell line, inhibited in the steady-state by PD-L1 expressed by APCs, here replaced by CHO expressing PD-L1, and the Raji B cell line also reporter (NF-κB) and expressing CD40. The biggest challenge of this project is to manage to deconvolve the luciferase signals from the 2 reporter lines because the test only measures a total luciferase signal without distinction, as it could be with fluorescence for example. This work and these data will be of great importance, in particular for the pharmaceutical industry faced with robust potency assays on increasingly complex molecules such as BiTEs. However, some important controls are missing and some discussion points would be welcome.

Comments on the methods, results and data interpretation

Importants comments and observations

-In the titration experiments using Jurkat cells in Figure 2A, an isotype control is missing. This is an essential control in order to guarantee the specificity of the bioassay. Indeed, in the absence of extensive phenotypic characterization of the Jurkat cells utilized in this study, we cannot exclude the expression on their surface of FcγR, e.g. CD16a, and therefore of non-specific receptors for the Fc part of antibodies and fusion constructs tested, with the cross-presentation in trans at the highest concentrations. The irrelevant antibody ‘mAb-X’ should be described a little more precisely, as to the isotype and if there is a fusion protein linked to the antibody part.
-Section 2.5 is more delicate because it involves being able to deconvolve the luciferase signals of Jurkat and Raji cells from a single luminescence measurement. First, the signals measured with Jurkat + CHO alone and with Raji alone, at the point of maximum concentration of anti-PD-L1/CD40L BsAbFP are of the order of 1e6 RLU and 7e5 RLU respectively. The addition of the two should make it possible to reach at least 1.7e6 RLU, which is not obtained in the assessment of the potency assay with a maximal luminescence of 1.2e6 RLU. There is then no synergistic effect as indicated in the title of the article, or even additive, and the phenomenon is all the more highlighted in the experiment shown in Figure 4. But this also raises the potential problem of signal saturation. Thus a study, with different Jurkat : CHO : Raji ratios, w/ or w/o luciferase reporter genes, is necessary.
-The parameters from the 4PL regression do not seem comparable between the experiments presented in Figures 3B / 4D, Figure 5A and Figure 6A, in particular the EC50 (closer to 1 ng/mL in Figure 6A), the upper asympotes greater than 2e6 in the last experiments, and the lower asymptotes. Please systematically present all these values in the text for each experiment, and specify if there are any modifications in the experimental design being studied, and if so which ones.
-The difference in duration between the PD-1, CD40L and dual-bioassay tests is not discussed. Indeed, the activation of the PD-1 signaling pathway requires an overnight time of 24 hours, unlike CD40L which requires a shorter incubation period of 4 h. How were these timing differences reconciled in the dual bioassay? This definitely adds to the complexity of having to measure the activation of 2 distinct signaling pathways in 2 different cellular systems, but with very different timing as well.

Minor points

-Title: there could be an ambiguity as to the “blockade of PD-1” interpretation when the authors generated a fusion protein with an anti-PD-L1 mAb. Indeed, the APCs also expressing PD-L2, another cognate ligand of PD-1 as described in the introduction, the construction will not be able to block PD-1 but rather the PD-1 / PD-L1 axis, as explained in section 2.1. Of course, this article is dedicated to the bioassay which focuses exclusively on PD-L1 and CD40, so the authors can see if this remark is relevant.
-Please indicate the standard deviations with the EC50 values obtained in sections 2.1 and 2.2.
-In Figure 4C, why not have tested the combination of Jurkat (NL), CHO and Raji (NL) instead of CHO + Raji (NL) only?
-In Figure 5A, why did you stop at a maximum concentration <1000 ng/mL when in previous experiments the maximum concentration was 3000 or 9000 ng/mL?
-In Figure 5A, the title of the x-axis should be “Antibody”

Round 2

Reviewer 2 Report

I thank the authors for taking into consideration my questions and suggestions made during my first round of review. The authors were able to answer all my questions, by adding clarifications and details in the revised version of the manuscript, or by providing data in the appendix (not shown to the reader), in particular regarding the potential presentation in trans, the titration data and the phenomenon of additivity/synergy discussed from the luciferase signals. I still have mixed feelings about the difference in the kinetics of PD-1 and CD40L signaling, but from my point of view this work is quite successful and will have a certain interest, especially in the context of the development of potency assays for the characterization of next generation immune checkpoint and costimulatory molecules, such as biTEs, for quality control processes.

Round 3

Reviewer 2 Report

na